# Knowledge Distillation in Wide Neural Networks: Risk Bound, Data Efficiency and Imperfect Teacher

**Guangda Ji**
Peking University
jiguangda@pku.edu.cn

**Zhanxing Zhu** ✉
School of Mathematical Sciences, Peking University
Center for Data Science, Peking University
zhanxing.zhu@pku.edu.cn

## Abstract

Knowledge distillation is a strategy of training a student network with guide of the soft output from a teacher network. It has been a successful method of model compression and knowledge transfer. However, currently knowledge distillation lacks a convincing theoretical understanding. On the other hand, recent finding on neural tangent kernel enables us to approximate a wide neural network with a linear model of the network's random features. In this paper, we theoretically analyze the knowledge distillation of a wide neural network. First we provide a transfer risk bound for the linearized model of the network. Then we propose a metric of the task's training difficulty, called data inefficiency. Based on this metric, we show that for a perfect teacher, a high ratio of teacher's soft labels can be beneficial. Finally, for the case of imperfect teacher, we find that hard labels can correct teacher's wrong prediction, which explains the practice of mixing hard and soft labels.

## 1 Introduction

Deep neural network has been a successful tool in many fields of artificial intelligence. However, we typically require deep and complex networks and much effort of training to achieve good generalization. Knowledge distillation(KD) is a method introduced in [9], which can transfer knowledge from a large trained model (i.e. *teacher*) to another smaller network (i.e. *student*). Through distillation, the student network can achieve better performance than direct training from scratch [9]. The vanilla form of KD in classification problem has a combined loss of soft and hard labels,

$$\mathcal{L} = \rho L(\mathbf{y}_s, \mathbf{y}_t) + (1 - \rho)L(\mathbf{y}_s, \mathbf{y}_g)$$

where $\mathbf{y}_t$ and $\mathbf{y}_s$ are *teacher* and *student*'s soft labels , $\mathbf{y}_g$ are *ground truth* labels and $\rho$ is called *soft ratio*.

Apart from this original form of KD, many variants that share the teacher-student paradigm are proposed. [10] uses intermediate layers of neural network to perform distillation. [23] and [25] adopt adversarial training to reduce the difference of label distribution between teacher and student. In [12] and [14], the authors consider graph-based distillation. Self distillation, proposed in [8], distills the student from an earlier generation of student of same architecture. The latest generation can outperform the first generation significantly.

A common conjecture on why KD works is that it provides extra information on the output distribution, and student model can use this "dark knowledge" to achieve higher accuracy. However, KD still lacks a convincing theoretical explanation. [21] argues that KD not only transfers super-class correlations to students, but also gives higher credits to correct samples in gradient flow. [2] finds that KD enables student to learn more task-related pixels and discard the background in image classification tasks.

[16] shows that self-distillation has a regularization effect on student's logits. However, very few works establish a comprehensive view on the knowledge distillation, including risk bound, the role of the soft ratio and how efficient the distillation makes use of data.

In this work, we attempt to deal with these issues with the help of neural tangent kernel and wide network linearization, i.e. considering distillation process for linearized neural networks. We focus on the soft ratio $\rho$ as it serves as a continuous switch between original hard label training and soft label distillation. The main contributions of our work are summarized as follows.

- We experimentally observe faster convergence rate of transfer risk with respect to sample size for softer tasks, i.e. with high $\rho$. We try to explain this with a new transfer risk bound for converged linearized student networks, based on distribution in random feature space. We show that the direction of weights converges faster for softer tasks. (Sec. 3)

- We introduce a metric on task's difficulty, called *data inefficiency*. Through this metric we show, for a perfect teacher, early stopping and higher soft ratio are beneficial in terms of making efficient use of data. (Sec. 4)

- We discuss the benefits of hard labels in imperfect teacher distillation in the scenario of KD practice. We show that a little portion of hard labels can correct student's outputs pointwisely, and also reduce the angle between student and oracle weight. (Sec. 5)

**Related Work**    Our work is built on neural tangent kernel techniques introduced in [11, 13]. They find that in the limit of infinitely wide network, the Gram matrix of network's random feature tends to a fixed limit called neural tangent kernel (NTK), and also stays almost constant during training. This results in an equivalence of training dynamics between the original network and linear model of network's random features. Therefore we replace the network with its linear model to avoid the trouble of nonlinearity. The most related work to ours is [18]. They consider distillation of linear models and gives a loose transfer risk bound. This bound is based on the probability distribution in feature space and therefore is different form the traditional generalization given by Rademacher complexity. We improve their bound and generalize their formulation to the case of linearization of an actual neural network.

## 2   Problem Setup

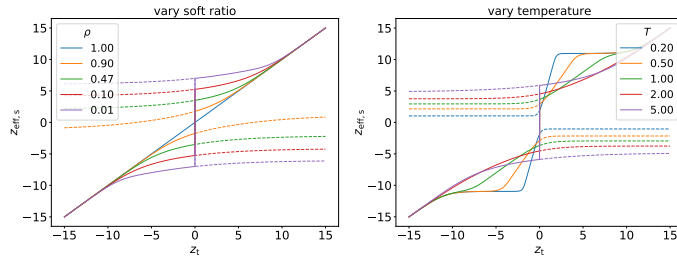

Figure 1: Effective student logits $z_{\text{s,eff}}$ as a function of $z_{\text{t}}$ and $y_{\text{g}}$. The **left** and **right** figure shows how soft ratio $\rho$ (with $T = 5.0$) and temperature $T$ (with $\rho = 0.05$) can affect the shape of $z_{\text{s,eff}}(z_{\text{t}}, y_{\text{g}})$. Each point is attained by solving Eq. 3 with first order gradient method. Solid lines show a correct teacher $y_{\text{g}} = \mathbb{1}\{z_{\text{t}} > 0\}$, and dashed lines denote a wrong teacher $y_{\text{g}} = \mathbb{1}\{z_{\text{t}} < 0\}$. The existence of hard label produces a discontinuity in $z_{\text{s,eff}}(z_{\text{t}}, y_{\text{g}})$.

We consider a binary classification problem on $x \in \mathcal{X} \subseteq \mathbb{R}^d$. We assume $x$ has a fixed distribution $P(x)$ and a ground truth classification boundary , $y = \mathbb{1}\{f_{\text{g}}(x) > 0\} \in \{0, 1\}$. The task is to find a student neural network $z = f(x; w) : \mathcal{X} \mapsto \mathbb{R}$ that best fits the data. $w$ are network's weights and $z$ is its output logit. The parameters are initialized with NTK-parameterization in [11],

$$h^1 = \sigma_W W^0 x / \sqrt{d} + \sigma_b b^0, \, x^1 = g(h^1),$$
$$h^{l+1} = \sigma_W W^l x^l / \sqrt{m} + \sigma_b b^l, \, x^{l+1} = g(h^{l+1}), \, l = 1, 2, \cdots, L-1, \qquad (1)$$
$$f(x; w) = \sigma_W W^L x^L / \sqrt{m} + \sigma_b b^L, \, w = \cup_{i=0}^{L} \{W^i, b^i\},$$

where $g(\cdot)$ is the activation function, $W_{ij}^l, b_i^l \sim \mathcal{N}(0, 1)$ are the parameters of each layer, and $(\sigma_W, \sigma_b)$ are the hyperparameters. $(\sigma_W, \sigma_b)$ are choose to be $(1.0, 1.0)$ throughout our experiments. The

network trained by empirical risk minimization $\hat{w} = \arg\min_w \mathcal{L}$ with (stochastic) gradient descent, and the *distillation loss* is

$$\mathcal{L} = \frac{1}{N}\sum_{n=1}^{N}\ell_n = \frac{1}{N}\sum_{n=1}^{N}\rho H(y_{\mathrm{t},n}, \sigma(\frac{z_{\mathrm{s},n}}{T})) + (1-\rho)H(y_{\mathrm{g},n}, \sigma(z_{\mathrm{s},n})), \tag{2}$$

where $H(p,q) = -[p\log q + (1-p)\log(1-q)]$ is binary cross-entropy loss, $\sigma(z) = 1/(1+\exp(-z))$ is sigmoid function, $z_{\mathrm{s},n} = f(x_n; w)$ are the output logits of *student* network, $y_{\mathrm{t},n} = \sigma(z_{\mathrm{t},n}/T)$ are soft labels of *teacher* network and $y_{\mathrm{g},n} = \mathbb{1}\{f_{\mathrm{g}}(x_n) > 0\}$ are the *ground truth* hard labels. $T$ is *temperature* and $\rho$ is *soft ratio*.

In this paper we focus on student's convergence behavior and neglect its training details. *We assume the student is over-parameterized and wide.* [7] proves the convergence of network to global minima for L2 loss under this assumption. We believe this holds true for distillation loss, and give a further discussion in Appendix (Sec. S1). This means that as training time $\tau \to \infty$, each student logit would converge to a target value that minimizes the loss of each sample,

$$\lim_{\tau\to\infty} z_{\mathrm{s}}(\tau) = \hat{z}_{\mathrm{s}}, \quad \frac{d\ell}{d\hat{z}_{\mathrm{s}}} = \frac{\rho}{T}(\sigma(\hat{z}_{\mathrm{s}}/T) - \sigma(z_{\mathrm{t}}/T)) + (1-\rho)(\sigma(\hat{z}_{\mathrm{s}}) - y_{\mathrm{g}}) = 0. \tag{3}$$

The implicit solution to this equation defines an *effective student logit* $z_{\mathrm{s,eff}} = \hat{z}_{\mathrm{s}}$ as a function of $z_{\mathrm{t}}$ and $y_{\mathrm{g}}$. In Fig. 1 we plot $z_{\mathrm{s,eff}}(z_{\mathrm{t}}, y_{\mathrm{g}})$ with varying soft ratio $\rho$ and temperature $T$. Solid curves show $z_{\mathrm{s,eff}}$ solved by Eq. 3 of a correct teacher $y_{\mathrm{g}} = \mathbb{1}\{z_{\mathrm{t}} > 0\}$, and dashed curves denote a wrong teacher. We can observe that the presence of hard label creates a split in the shape of $z_{\mathrm{s,eff}}$, and this split increases as hard label takes higher ratio. The generalization and optimization effect of this split will be discussed in Sec. 3 and 4.

The temperature $T$ is introduced by Hilton in [9] to soften teacher's probability. Intuitively $\sigma'(z) \to 0$ when $\sigma(z) \to \{0,1\}$, so a higher $T$ makes student easily converged. However, this is only a benefit during training. In Fig. 1 we show that converged student logits are always bigger than their teachers', $|z_{\mathrm{s,eff}}| > |z_{\mathrm{t}}|$. We also observed that when $T > 1$, a higher temperature causes the split to be wider, and when $T < 1$, the curve forms a $T$-independent plateau. In this paper we are interested in the effect of soft ratio $\rho$. Therefore, we set temperature to be fixed and follow the convention $T > 1$.

Wide network assumption also allows us to linearize student network with NTK techniques. According to [11, 13], the outputs of an infinitely wide network are approximately its linearization at initialization, $f(x; w_{\mathrm{nlin}}) \approx f(x; w_0) + \Delta_w^\top \phi(x)$, where $\phi(x) = \partial_w f(x; w_0) \in \mathbb{R}^p$ are called *random features*, $p$ is the dimension of weight, $w_{\mathrm{nlin}}$ is the weight of original network and $\Delta_w = w - w_0 \in \mathbb{R}^p$ is the *weight change* of linear model trained by same strategy as the original network. The converged weight change $\Delta_{\hat{w}}$ is used throughout this paper. For training samples of $\mathbf{X} = [x_1, \cdots, x_n] \in \mathbb{R}^{d\times n}$ and $\mathbf{z} = [z_1, \cdots, z_n]^\top \in \mathbb{R}^{n\times 1}$, the converged weight change is

$$\Delta_{\hat{w}} = \phi(\mathbf{X})(\hat{\Theta}(\mathbf{X}, \mathbf{X}))^{-1}\Delta_{\mathbf{z}}, \tag{4}$$

where $\hat{\Theta}(\mathbf{X}, \mathbf{X}) = \phi(\mathbf{X})^\top \phi(\mathbf{X})$ is called the empirical kernel, $\Delta_{\mathbf{z}} = \mathbf{z} - \mathbf{z}_0$ and $\mathbf{z}_0 = f(\mathbf{X}; w_0)$ are student's logits at initialization. According to [11, 13], in infinite width limit, $\hat{\Theta}(\mathbf{X}, \mathbf{X})$ converges to a weight independent kernel $\Theta(\mathbf{X}, \mathbf{X})$, called NTK. We assume $\hat{\Theta}(\mathbf{X}, \mathbf{X}) \approx \Theta(\mathbf{X}, \mathbf{X})$ throughout this paper and simplify it as $\Theta_n$. Eq. 4 can be proved by Theorem 1 of [18]. Eq.8 of [13] also gives a similar result of $\ell_2$-loss. Note that due to the extremely high dimension of $\phi(\mathbf{X})$, direct calculation of Eq. 4 is impractical. We can instead attain $\Delta_{\hat{w}}$ by training the linear model.

The rest of this paper are organized as follows: In Sec. 3 we give our transfer risk bound of linearized network. This bound is computationally expensive, so in Sec. 4 we introduce a more direct metric, called data inefficiency. Then we analyze the effect of teacher's early stopping and soft ratio with this metric. Sec. 3 and 4 only consider a perfect teacher and under this setting, hard labels are not beneficial for KD. Therefore as a complementary, we study the effect of hard labels in imperfect teacher distillation in Sec. 5.

## 3 Transfer Risk Bound

The *transfer risk* is the probability of different prediction from teacher, $\mathcal{R} = \mathbb{P}_{x\sim P(x)}[z_{\mathrm{t}} \cdot z_{\mathrm{s}} < 0]$.

Before we state our bound, we first present our observation on perfect teacher distillation (Fig. 2

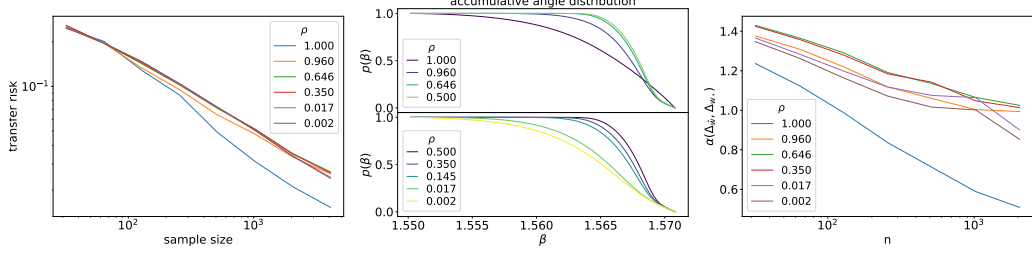

Figure 2: **Left:** Experimental transfer risk, plotted with respect to sample size $n$. The curve shows a power law relation, with a faster rate for pure soft distillation. **Middle:** Accumulative angle distribution $p(\beta)$, as part of our transfer risk bound. We split the curves into two subfigures because they change non-monotonically with respect to $\rho$. **Right:** $\alpha'_n = \alpha(\Delta_{\hat{w}}, \Delta_{w_*})$ with respect to $n$. See Sec. S4 in Appendix for details.

left). $\mathcal{R}$ shows a power relation with respect to sample size $n$, and pure soft distillation shows a significantly faster converging rate. This power law is also observed in practical distillation of ResNet on CIFAR (see Sec. S2 in Appendix). The relation of $\mathcal{R}$ and $n$ is rarely discussed in deep learning. However in statistical learning, [22, 15] prove that given some criterion on the logits distribution near class boundary, this rate can approach $O(1/n)$, instead of $O(1/\sqrt{n})$ given by traditional VC theory. Our results show experimental connections to their claim. We try to explain this with our transfer risk bound. Compared with the bound in [18] for linear distillation, our bound is a tightened version of their results and a modification to linearized network, as elaborated in the following.

We assume the student is expressive enough to approximate effective logits and zero function. Then $z_{\text{s,eff}} \approx f(x; w_0) + \Delta_{w_*}^\top \phi(x)$ and $0 \approx f(x; w_0) + \Delta_{w_z}^\top \phi(x)$ where $\Delta_{w_*}$ is *oracle weight change* and $\Delta_{w_z}$ is *zero weight change*. This approximation relies on the expressiveness of student, and has no dependency on teacher's weight. The student can be simplified as a bias-free linear model, $z \approx (\Delta_w - \Delta_{w_z})^\top \phi(x)$ and transfer risk is $\mathcal{R} = \mathbb{P}_{x \sim P(x)} [(\Delta_{w_*} - \Delta_{w_z})^\top \phi(x) \cdot (\Delta_{\hat{w}} - \Delta_{w_z})^\top \phi(x) < 0]$.

Further, student's weight change can be written as a projection onto $\text{span}\{\phi(\mathbf{X})\}$,

$$\Delta_{\hat{w}} = \phi(\mathbf{X})\Theta_n^{-1}\phi(\mathbf{X})^\top \Delta_{w_*} = \mathbf{P}_\Phi \Delta_{w_*}, \tag{5}$$

where $\mathbf{P}_\Phi$ is the projection matrix. We denote $\bar{\alpha}(a, b) = \cos^{-1}(|x^\top y|/\sqrt{x^\top x \cdot y^\top y}) \in [0, \pi/2]$ as the angle of two vectors $a, b$. Acute angle is used so that wrong prediction of both classes is counted. Similar to [18], the *accumulative angle distribution* $p(\beta)$ of given distribution is defined as

$$p(\beta) = \mathbb{P}_{x \sim P(x)} [\bar{\alpha}(\phi(x), \Delta_{w_*} - \Delta_{w_z}) > \beta], \text{ for } \beta \in [0, \pi/2]. \tag{6}$$

Now we state our result of transfer risk. The proof is in Sec. S3 of Appendix.

**Theorem 1.** *(Risk bound) Given input distribution $P(x)$, training samples $\mathbf{X} = [x_1, \cdots, x_n]$, oracle weight change $\Delta_{w_*}$, zero weight change $\Delta_{w_z}$ and accumulative angle distribution $p(\beta)$, the transfer risk is bounded by,*

$$\mathcal{R}_n \le p(\frac{\pi}{2} - \bar{\alpha}_n), \tag{7}$$

*where $\bar{\alpha}_n = \bar{\alpha}(\Delta_{w_*} - \Delta_{w_z}, \Delta_{\hat{w}} - \Delta_{w_z})$ and $\Delta_{\hat{w}}$ is student's converged weight change.*

### 3.1 Estimation of the Bound

The $p(\beta)$ and $\bar{\alpha}_n$ in Theorem. 1 can be estimated with NTK techniques. For $p(\beta)$, calculation involves sampling over $\cos \bar{\alpha}(\phi(x), \Delta_{w_*} - \Delta_{w_z})$, and

$$\cos \bar{\alpha}(\phi(x), \Delta_{w_*} - \Delta_{w_z}) = \frac{(\Delta_{w_*} - \Delta_{w_z})^T \phi(x)}{||\Delta_{w_*} - \Delta_{w_z}||_2 \cdot ||\phi(x)||_2} = \frac{z_{\text{s,eff}}(x)}{||\Delta_{w_*} - \Delta_{w_z}||_2 \cdot \sqrt{\Theta(x, x)}}. \tag{8}$$

$\Delta_{w_*}$ and $\Delta_{w_z}$ can be approximated by online-batch SGD training, which is equivalent to training with infinite amount of samples. Fig. 2 middle shows estimations of $p(\beta)$ of this method. It shows a non-monotonicity with $\rho$, but all curves shows a near linear relation with $\beta$ near $\beta \to \pi/2$.

For $\bar{\alpha}_n$, training an actual student is approachable, but we can also approximate it beforehand with a straightforward view. Usually, zero function is much easier to converge than a normal function, and $||\Delta_{w_z}||_2 \ll ||\Delta_{w_*}||_2$ (see Sec. S5 in Appendix). Then,

$$\cos \bar{\alpha}_n \approx \cos \bar{\alpha}(\Delta_{w_*}, \Delta_{\hat{w}}) = \frac{\Delta_{w_*}^\top \mathbf{P}_{\mathbf{\Phi}} \Delta_{w_*}}{||\Delta_{w_*}||_2 ||\mathbf{P}_{\mathbf{\Phi}} \Delta_{w_*}||_2} = \frac{\sqrt{\Delta_{\mathbf{z}}^\top \Theta_n^{-1} \Delta_{\mathbf{z}}}}{||\Delta_{w_*}||_2} = \frac{||\Delta_{\hat{w}}||_2}{||\Delta_{w_*}||_2}. \tag{9}$$

Take $\rho = 1$ as an example. From Fig. 4 right we empirically observe that $\partial \ln ||\Delta_{\hat{w}}||_2 / \partial \ln n \sim n^{-b}$, where $b$ is the parameter to describe this relation. By integrating over $n$, we can get an asymptotic behavior of $\alpha_n \sim n^{-b/2}$. Then based on the near linearity of $p(\beta)$ near $\beta \to \pi/2$ our result gives a bound of $O(n^{-b/2})$. When $b > 1$, this bound outperforms classical bounds. However, we are not certain whether this is the case since $b$ depends on various hyper-parameters, but we do find $b$ to be bigger when teacher's stopping epoch is small (i.e. the task is easy). Note that this estimation requires the existence of a converging $||\Delta_{\hat{w}}||_2$ with respect to $n$. These assumptions fail to be satisfied when hard labels are added to the loss (Fig. 4 right). As a complement, in Fig. 2 right we plot the direct calculation of $\alpha_n$ of different soft ratio. The result shows the fastest convergence speed in pure soft distillation case. This explains the result of Fig. 2 left about the fast convergence with pure soft labels.

## 3.2 Tightness Analysis

First we show that the risk bound in [18] is quite loose in high dimensions. Both their results and ours use a property that $\bar{\alpha}'_n = \alpha(\Delta_{\hat{w}}, \Delta_{w_*})$ monotonically decreases with respect to sample size $n$ (Lemma.1 in [18]). However, they utilize this property loosely that $\bar{\alpha}'_n$ is approximated by $\alpha$ trained by one sample, $\bar{\alpha}'_n \le \bar{\alpha}'_1 = \bar{\alpha}(\Delta_{w_*}, \phi(x_i))$. This leads to a risk bound of $R_n \le \min_\beta p(\beta) + p(\pi/2 - \beta)^n$. Due to the high dimension of random vector $\phi(x)$, $\phi(x)$ and $\Delta_{w_*} - \Delta_{w_z}$ are very likely to be perpendicular (Fig. 2 middle). We can further show (see Sec. S6 in Appendix) that for ReLU activation, $p(\beta) \equiv 1$ strictly for $\beta \in [0, \beta_t], \beta_t \approx \pi/2$ and therefore their bound is strictly $\mathcal{R} \equiv 1$. We tighten their result by directly using $\alpha'_n$ in risk estimation. The improvement is significant since even if $\bar{\alpha}'_1 \approx \pi/2$, $\bar{\alpha}'_n$ can be small when $n$ is sufficiently large. However, we have to point out that our bound also shows certain looseness with small sample size due to the fact that $p(\pi/2 - \alpha_{n_{\text{small}}}) \approx 1$. The generalization ability of small sample size remains mysterious.

Our approach and [18] differ from classic learning theory for generalization bound of neural network [17, 4, 1]. They are based on Rademacher complexity $\mathfrak{R}$ of hypothesis space $\mathcal{H}$, and give a bound like $L_{\mathcal{D},0-1}(f) \le 2\mathfrak{R}_n(\ell \circ \mathcal{H}) + O(\sqrt{\ln(1/\delta)/n})$ where $\mathfrak{R}_n(\mathcal{H}) = \mathbb{E}_{\xi \sim \{\pm 1\}^n} \left[ \sup_{f \in \mathcal{H}} \sum_{i=1}^n \xi_i f(x_i) \right] / n$. A common way to tighten this bound is to restrict $\mathcal{H}$ near the network's initialization, $\mathcal{H} = \{f(x; w) | w \in \mathbb{R}^p, \text{s.t.} ||w - w_0||_2 \le B\}$ ([1, 3, 4]). However as [26] shows, the generalization error is highly task related. Even if $\mathcal{H}$ is restricted, Rademacher complexity still only captures the generalization ability of the hardest function in $\mathcal{H}$. Our approach, instead, tackles directly the the task related network function $f(x) = f(x; w_0) + \Delta_{w_*}^\top \phi(x)$. Therefore the hypothesis space of our bound is much smaller.

# 4  Data Inefficiency Analysis

In the discussion above, $||\Delta_{\hat{w}}||_2$ plays an important role in the calculation of $p(\beta)$ and angle convergence. However, $p(\beta)$ needs much effort to calculate and cannot show the obvious advantage of soft labels. In this section, we define a metric on the task's training difficulty, called *data inefficiency* to directly measure the change of $||\Delta_{\hat{w}}||_2$. We first state its rigorous definition and then discuss how the stopping epoch of teacher and soft ratio affect data inefficiency.

**Definition 1.** *(Data Inefficiency) We introduce data inefficiency as a discrete form of* $\partial \ln ||\Delta_{\hat{w},n}||_2 / \partial \ln n$,

$$\mathcal{I}(n) = n \left[ \ln \mathbb{E} ||\Delta_{\hat{w},n+1}||_2 - \ln \mathbb{E} ||\Delta_{\hat{w},n}||_2 \right] \tag{10}$$

*where* $||\Delta_{\hat{w},n}||_2 = \sqrt{\Delta_{\mathbf{z}_n}^\top \Theta_n^{-1} \Delta_{\mathbf{z}_n}}$ *is the norm of student's converged weight change trained by $n$ samples.*

The expectation is taken over the randomness of samples and student's initialization. Logarithm is used to normalize the scale of $\Delta_z$, and to reveal the power relation of $||\Delta_{\hat{w}}(n)||_2$. We define data inefficiency as Eq. 10 for the reasons using the following principle.

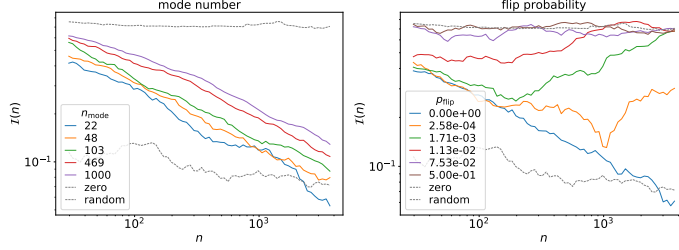

Figure 3: **Left:** Difficulty control on the number of modes. The figure shows $\mathcal{I}(n)$ of learning different Gaussian mixture function. The decreasing behavior of $\mathcal{I}(n)$ is typical for learning a noise-free smooth function. **Right:** Difficulty control on flip probability. The figure plots $\mathcal{I}(n)$ of learning the same function with different noise level. $p_{\text{flip}} = 0.5$ means a completely random sign. The noise makes these tasks so difficult to learn that $\mathcal{I}(n) \equiv 0.8$, this means $\Delta_{\hat{w}}$ will not converge. The two figures demonstrate a positive correlation between $\mathcal{I}(n)$ and task's difficulty. The dashed lines are references of a hard and easy task. The upper dashed line shows the complexity of random label $\Delta_z \sim \mathcal{N}(0, 1)$, while the lower dashed line shows the complexity of zero function $z \equiv 0$. The later one also demonstrates that zero function is extremely easy to learn and $\Delta_{w_z}$ can be neglected. All the results are based on the average of 20 runs.

**Principle 1.** $\mathcal{I}(n)$ *reveals the student's difficulty of weight recovery from teacher.*

For better explanation, we assume again there exists an oracle weight change $\Delta_{w_*}$ as we did in Sec. 3. Then student's weight change is a projection, $\Delta_{\hat{w}} = \mathbf{P}_\Phi \Delta_{w_*}$, where $\mathbf{P}_\Phi$ is a projection matrix onto $\text{span}\{\phi(\mathbf{X})\}$. As $n$ increases and $\text{span}\{\phi(\mathbf{X})\}$ expands, $\Delta_{\hat{w}}$ gradually recovers to $\Delta_{w_*}$, and $||\Delta_{\hat{w}}||_2 = \sqrt{\Delta_{\hat{w}}^\top \mathbf{P}_\Phi \Delta_{\hat{w}}}$ shows the stage of recovery. If the task is data efficient, we can recover the main component of $\Delta_{w_*}$ with a small sample size, and further we expect $||\Delta_{\hat{w}}||_2$ not to increase very much with $n$. Reversely, if the task is inefficient, the same sample set is not sufficient to recover the main component of $\Delta_{w_*}$, so we expect $||\Delta_{\hat{w}}||_2$ to continue increasing. Therefore, we use $\mathcal{I}(n)$ to indicate the increasing speed of $||\Delta_{\hat{w},n}||_2$ with respect to $n$ and a faster increasing (or slower converging) $\mathcal{I}(n)$ indicates a less data efficient task to train.

To demonstrate this principle, we perform two difficulty control tasks of learning Gaussian mixture function $z_{\text{gaussian}}(x) = \sum_{j=1}^q A_j \exp(-(x - x_j)^2/\sigma_j^2)$ (see Sec. S4 in Appendix for details). The difficulty of the first task is controlled by the number of Gaussian modes $q$ (Fig. 3 left). In the second task (Fig. 3 right), we control difficulty by the probability of sign flipping $z_* = s \times z_{\text{gaussian}}(x)$, where $s$ has probability of $\{1 - p_{\text{flip}}, p_{\text{flip}}\}$ to be $\{1, -1\}$. Both experiments show that $\mathcal{I}(n)$ ranges as the order of difficulties, which agrees with our idea on data inefficiency.

Our idea is also supported by other works. [1] proves that for 2-layer over-parameterized network trained with $\ell_2$-loss, $\sqrt{\Delta_{\mathbf{z}}^\top \Theta_n^{-1} \Delta_{\mathbf{z}}/n}$ is a generalization bound for the global minima. [3] studies $L$-layer over-parameterized network trained by online-SGD and get a similar bound of $\widetilde{\mathcal{O}}[L \cdot \sqrt{\Delta_{\mathbf{z}}^\top \Theta_n^{-1} \Delta_{\mathbf{z}}/n}]$. A slower increasing $\sqrt{\Delta_{\mathbf{z}}^\top \Theta_n^{-1} \Delta_{\mathbf{z}}}$ means a faster decreasing generalization error, which also means the task is more data efficient. [19] supports our idea from the perspective of optimization. They calculate $\Theta(x, x)$'s eigensystem in the case of uniform distribution on $S^d$ and find that convergence time of learning the $i$th eigen function with $\ell_2$-loss is proportional to $\lambda_i(\Theta)^{-1} \propto \mathbf{z}^\top \Theta_n^{-1} \mathbf{z}$.

## 4.1 Early Stopping Improves Data Efficiency

It is a well-known result that neural network first learns the low frequency components of the target function [24]. Therefore, early stopping the teacher provides the student with a simplified version of the target function, which may be easier to train. For the integrity of this work, we show in Fig. 4 middle that early stopping improves the data efficiency.

[5] experimentally observes that a more accurate teacher may not teach a better student, and early stopping the teacher helps the student to fit the data better. A recent work [6] also emphasizes the importance of early stopping in KD theoretically. They assume a probability of data corruption and

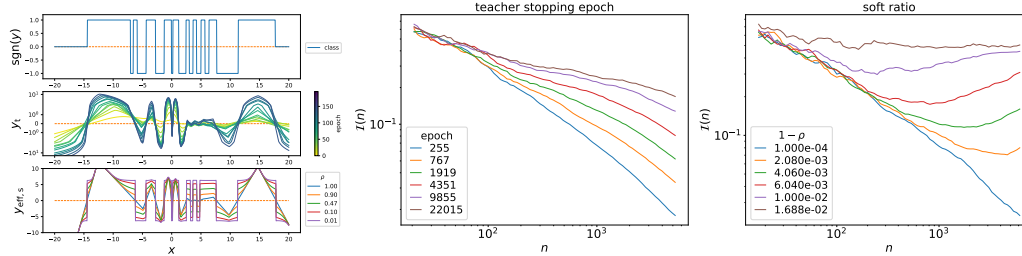

Figure 4: **Left:** A 1-D example of teacher and student output. **Left Top**: Ground truth class boundary. **Left Middle**: Teacher's logits at different stopping epoch. The scale of teacher increases and shape becomes more detailed while training. **Left Bottom**: Effective student logits $y_{s,eff}$ at different soft ratio $\rho$. This figure further illustrates the discontinuity in $y_{s,eff}$. For a small $\rho$, students shape shows a clear similarity with label smoothing. All student share a same teacher network. **Middle and Right:** Data inefficiency curve of different teacher stopping epoch and soft ratio. It shows that adding hard labels to distillation increases sample complexity. See Sec. S4 for details.

treat the high frequency component in teacher's label as a result of noise. They claim that early stopping prevents teacher from learning the noise in its label.

## 4.2 Hard labels Increase Data Inefficiency

In this section we study the splitting effect of training with adding hard labels. Fig. 4 right shows $\mathcal{I}(n)$ with respect to different soft ratio $\rho$. It also shows a transition from a decreasing curve to constant $\mathcal{I}(n) \approx 0.5$. This demonstrates that hard labels increase data inefficiency. This can be explained that discontinuity increases the higher frequency component of the $z_{s,eff}(x)$'s shape, so the shape becomes harder to fit. The constant of $\mathcal{I}(n)$ in pure hard label training is lower than that of total random logits (Fig. 3). This suggests that learning a discontinuous function is still easier than a random function.

**Connection with label smoothing.** KD resembles *label smoothing* (LS) technique introduced in [20]. LS can be regarded as replacing a real teacher with a uniform distribution $P_k = 1/K$ and $y_{LS} = \epsilon/K + (1 - \epsilon)y_g$, in order to limit student's prediction confidence without the effort of training a teacher. In our setting of $K = 2$, $z_{LS,eff} = a \cdot \text{sgn}(y_g)$, where $a = \log(2/\epsilon - 1)$. In KD, similarly, student effective logit is approximately a cut-off function (Fig. 4 left bottom) $z_{s,eff} \approx \text{sgn}(z_t) \cdot \max\{t, |z_t|\}$, where the threshold is $t = z_{s,eff}(0_+, 1)$. As the soft ratio $\rho \to 0$, $t$ exceeds $\max\{|z_t|\}$ and student's logits tend to a sign function $z_{s,eff} \to t \cdot \text{sgn}(z_g)$. Therefore in this limit KD tends to be LS. Results in Fig. 4 right suggest that LS is not a easy task compared with pure soft KD. We have to point out the difference between two methods. LS is introduced to upper limit student's logits $\max\{|z_{LS}|\} < a$, while on the contrary, in KD, student's logits are lower limited by a threshold, as we see in Fig. 4 left bottom.

To conclude, the benefit of KD is that teacher provides student with smoothened output function that is easier to train than with hard labels. This function is easier to fit when teacher is earlier stopped, but at a cost of losing details on class boundary. Adding hard labels damages the smoothness and therefore makes the task more difficult.

## 5 Case of Imperfect Teacher

We emphasize that in previous discussion the teacher is assumed to be perfect and teacher's mistake is not considered. Sec. 3 discusses transfer risk instead of generalization error and in Sec. 4 we study the difficulty of fitting teacher's output. In both sections the result shows a favor of pure soft label distillation. However, according to empirical observations in the original paper of KD [9], a small portion of hard labels are required to achieve better generalization error than pure soft ones. KD on practical datasets and network structure (Fig. 5 left) also demonstrates this claim. In this section we analyze benefit of hard label in imperfect teacher distillation.

**Hard labels correct student's logits at each point.** The dashed lines in Fig. 1 left shows $z_{s,eff}$ when teacher makes a wrong prediction $y_g \neq \mathbb{1}\{z_t > 0\}$. From the figure we can see that $z_{s,eff}$ moves closer to the correct direction. To be more precise, we take $y_g = 1, T = 1.0$ case as an example and solve $z_{s,eff}$ according to Eq. 3. No matter how wrong the teacher is, the existence of hard label

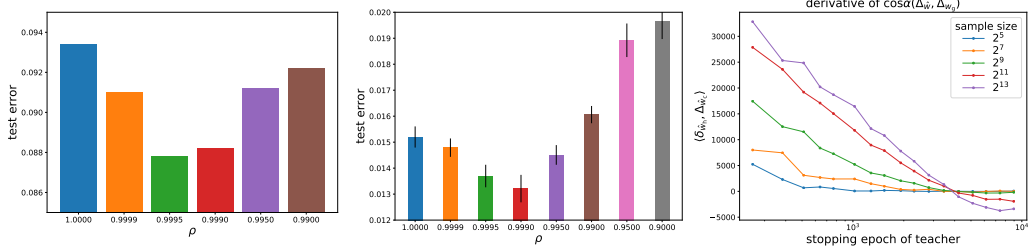

Figure 5:    **Left and Middle:** Imperfect distillation on synthetic dataset and practical (CI-FAR10/ResNet) dataset. These plots show that pure soft label distillation in imperfect KD is not optimal. **Right:** $\langle \delta_{\hat{w}_\mathrm{h}}, \Delta_{\hat{w}_\mathrm{c}} \rangle$ is proportional to $\partial \cos \alpha(\Delta_{\hat{w}}, \Delta_{w_\mathrm{g}})/\partial(1-\rho)$. The sign of it denotes whether adding hard labels can or cannot reduce the angle between the student and oracle. The stopping epoch of teacher is positively related to teacher's generalization ability. The epoch when $\langle \delta_{\hat{w}_\mathrm{h}}, \Delta_{\hat{w}_\mathrm{c}} \rangle$ switches sign, is approximately when teacher outperforms the best of student network. See Sec. S4 for details.

restrict the wrongness of student to $\sigma(z_{\mathrm{s,eff}}) \geq 1 - \rho$. Further, if $\rho \leq 1/2$, $z_{\mathrm{s,eff}}$ is always correct. Therefore hard labels can pointwisely alleviate the wrongness of student's logits.

**Hard labels guide student's weight to correct direction.** In this section we again assume three oracle weight change of ground truth $\Delta_{w_\mathrm{g}}$, teacher $\Delta_{w_\mathrm{t}}$ and student $\Delta_{w_\mathrm{s,eff}}$ to express $z_\mathrm{g}, z_\mathrm{t}$ and $z_{\mathrm{s,eff}}(z_\mathrm{t}, \mathrm{sgn}(z_\mathrm{g}))$ (We switch the expression $\Delta_{w_*}$ to $\Delta_{w_\mathrm{g}}$ to better distinguish ground truth, teacher and student). Then student learns a projected weight change of $\Delta_{\hat{w}} = \mathbf{P}_\Phi \Delta_{w_\mathrm{s,eff}}$. For simplicity, we denote $\mathbf{a}^\top \Theta_n^{-1} \mathbf{b} = \langle \mathbf{a}, \mathbf{b} \rangle_{\Theta_n}$ and $\mathbf{a}^\top \mathbf{b} = \langle \mathbf{a}, \mathbf{b} \rangle$.

As we show in Sec. 3, student with a smaller $\alpha(\Delta_{\hat{w}}, \Delta_{w_g})$ generalizes better. Therefore we consider the cosine of student and oracle weight change,

$$\cos \alpha(\Delta_{\hat{w}}, \Delta_{w_\mathrm{g}}) = \frac{\Delta_{w_\mathrm{g}}^\top \Delta_{\hat{w}}}{||\Delta_{w_\mathrm{g}}||_2 \sqrt{\Delta_{\hat{w}}^\top \Delta_{\hat{w}}}} = \frac{\langle \Delta_{\mathbf{z}_\mathrm{g}}, \Delta_{\mathbf{z}_\mathrm{s,eff}} \rangle_{\Theta_n}}{||\Delta_{w_\mathrm{g}}||_2 \sqrt{\langle \Delta_{\mathbf{z}_\mathrm{s,eff}}, \Delta_{\mathbf{z}_\mathrm{s,eff}} \rangle_{\Theta_n}}}. \tag{11}$$

If hard labels are beneficial, then we expect $\cos \alpha(\Delta_{\hat{w}}, \Delta_{w_\mathrm{g}})$ to increases with respect to $1 - \rho$. The dependency of $\Delta_{w_\mathrm{s,eff}}$ on $\Delta_{w_\mathrm{g}}$ and $\Delta_{w_\mathrm{t}}$ is implicit, so we only consider the effect of hard labels near pure soft distillation ($\rho \to 1$) from an imperfect teacher. Then the change of $\cos \alpha(\Delta_{\hat{w}}, \Delta_{w_\mathrm{g}})$ with respect to hard label can be approximated by linear expansion, as summarized by the following theorem.

**Theorem 2.** *(Effect of Hard Labels)* We introduce correction logits $\delta z_\mathrm{h}$ to approximate $z_{\mathrm{s,eff}}$ solved by Eq. 3 as a linear combination $z_{\mathrm{s,eff}} \approx z_\mathrm{t} + (1-\rho)\delta z_\mathrm{h}$ in the limit of $\rho \to 1$. Then the derivative of $\cos \alpha(\Delta_{\hat{w}}, \Delta_{w_\mathrm{g}})$ with respect to hard ratio $1 - \rho$ is,

$$\begin{aligned}
\left. \frac{\partial \cos \alpha(\Delta_{\hat{w}}, \Delta_{w_\mathrm{g}})}{\partial(1-\rho)} \right|_{\rho=1} = & \frac{1}{||\Delta_{w_\mathrm{g}}||_2 \sqrt{\langle \Delta_{\mathbf{z}_\mathrm{t}}, \Delta_{\mathbf{z}_\mathrm{t}} \rangle_{\Theta_n}}} \times \\
& \left( \langle \Delta_{\mathbf{z}_\mathrm{g}}, \delta \mathbf{z}_\mathrm{h} \rangle_{\Theta_n} - \frac{\langle \Delta_{\mathbf{z}_\mathrm{g}}, \Delta_{\mathbf{z}_\mathrm{t}} \rangle_{\Theta_n}}{\langle \Delta_{\mathbf{z}_\mathrm{t}}, \Delta_{\mathbf{z}_\mathrm{t}} \rangle_{\Theta_n}} \langle \Delta_{\mathbf{z}_\mathrm{t}}, \delta \mathbf{z}_\mathrm{h} \rangle_{\Theta_n} \right).
\end{aligned} \tag{12}$$

The sign of this derivative indicates whether hard label benefits or not. The expression in the parentheses has an intuitive geometric meaning. It can be written as a projection $\langle \delta_{\hat{w}_\mathrm{h}}, \Delta_{\hat{w}_\mathrm{c}} \rangle$, where $\delta_{\hat{w}_\mathrm{h}} = \phi(\mathbf{X})\Theta_n^{-1}\delta z_\mathrm{h}$ is the change of student's weight by adding hard labels, and $\Delta_{\hat{w}_\mathrm{c}} = \Delta_{\hat{w}_\mathrm{g}} - \Delta_{\hat{w}_\mathrm{t}} \frac{\langle \Delta_{\hat{w}_\mathrm{t}}, \Delta_{\hat{w}_\mathrm{g}} \rangle}{\langle \Delta_{\hat{w}_\mathrm{t}}, \Delta_{\hat{w}_\mathrm{t}} \rangle}$ is the orthogonal complement of $\Delta_{\hat{w}_\mathrm{t}} = \phi(\mathbf{X})\Theta_n^{-1}\Delta \mathbf{z}_\mathrm{t}$ with respect to $\Delta_{\hat{w}_\mathrm{g}} = \phi(\mathbf{X})\Theta_n^{-1}\Delta \mathbf{z}_\mathrm{g}$. $\Delta_{\hat{w}_\mathrm{c}}$ tells the correct direction where the student can improve most. Therefore, the sign of this projection $\langle \delta_{\hat{w}_\mathrm{h}}, \Delta_{\hat{w}_\mathrm{c}} \rangle$ shows whether hard labels can lead $\Delta_{\hat{w}}$ to or against the correct direction.

Fig. 5 right plots $\langle \delta_{\hat{w}_\mathrm{h}}, \Delta_{\hat{w}_\mathrm{c}} \rangle$ with respect to teacher's stopping epoch $e$, which is positively related to teacher's generalization ability. Interestingly, when $e$ is small and teacher is not perfect, $\langle \delta_{\hat{w}_\mathrm{h}}, \Delta_{\hat{w}_\mathrm{c}} \rangle$ is positive for all sample sizes, but when $e$ is big enough where the teacher outperforms the best of hard label training student, $\langle \delta_{\hat{w}_\mathrm{h}}, \Delta_{\hat{w}_\mathrm{c}} \rangle$ becomes negative, which means teacher is so accurate that

hard labels cannot give useful hint. As a short conclusion, adding hard labels is a balance between the inefficiency of data usage, and the correction from teacher's mistakes.

# 6 Conclusion

In this work, we attempt to explain knowledge distillation in the setting of wide network linearization. We give a transfer risk bound based on angle distribution in random feature space. Then we show that, the fast decay of transfer risk for pure soft label distillation, may be caused by the fast decay of student's weight angle with respect to that of oracle model. Then we show that, early stopping of teacher and distillation with a higher soft ratio are both beneficial in making efficient use of data. Finally, even if hard labels are data inefficient, we demonstrate that they can correct an imperfect teacher's mistakes, and therefore a little portion of hard labels are needed in practical distillation.

For future work, we would like to design new form of distillation loss which does not suffer from discontinuity and at the same time, can still correct teacher's mistakes. We would also like to tighten our transfer risk bound and fit it to a practical nonlinear neural network.

# Broader Impact

Knowledge distillation is a heavily used technique for model compression. It is of high industrial importance since small models are easy deployed and cost saving. Unfortunately, it remains as a black box. The lack of a theoretical explanation limits a wider application of this technique. Our work provide a theoretical analysis and rethinking of this technique. Even though this work has no direct societal impact, with the guidance of our analysis, new distillation techniques of higher efficiency and performance may be proposed. However, development of model compression techniques can have negative impact: it reduces the time and cost of training functional deep learning applications, which make the abuse of deep learning techniques easier. We suggest that deep learning community and governments to put forward stronger restrictions on the opensource of data and models, and more careful supervision on the abuse of computing power.

# Acknowledgement

This project is supported by The National Defense Basic Scientific Research Project, China (No. JCKY2018204C004), National Natural Science Foundation of China (No.61806009 and 61932001), PKU-Baidu Funding 2019BD005, Beijing Academy of Artificial Intelligence (BAAI).

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
