[Supplementary Material · nips_distillation_paper.pdf]

# Supplementary for Knowledge Distillation in Wide Neural Networks: Risk Bound, Data Efficiency and Imperfect Teacher

## S1   Convergence of Distillation Loss

Even though [1] only proves the convergence for only L2 loss, we believe this also holds for our distillation loss, with a little bit of modification. The key idea of [1] is that, convergence is guaranteed by the near constancy of NTK matrix $\hat{\Theta}(\mathbf{X}, \mathbf{X})$. Then we can use the following equation to prove convergence,

$$\dot{\mathbf{z}}_{\mathrm{s}} = -\eta \hat{\Theta}(\mathbf{X}, \mathbf{X})(\mathbf{z}_{\mathrm{s}} - \mathbf{z}_{\mathrm{eff}}). \tag{S1}$$

As proved in the original paper of NTK([2]), the near constancy of $\hat{\Theta}(\mathbf{X}, \mathbf{X})$ has no requirement on the type of loss, so this is also true for our distillation loss. Then, the difference only lies in the second term $\mathbf{z}_{\mathrm{s}} - \mathbf{z}_{\mathrm{eff}}$, which in the case of distillation, is substituted with $\partial_{\mathbf{z}_{\mathrm{s}}} \mathcal{L}(\mathbf{z}_{\mathrm{s}}, \mathbf{z}_{\mathrm{eff}})$ (same as Eq. 6 in [3]). Due to the fact of finite training data and the convexity of $\mathcal{L}$ (w.r.t $\mathbf{z}_{\mathrm{s}}$), the gradient can be lower bounded by another L2 loss, therefore

$$|\partial_{\mathbf{z}_{\mathrm{s}}} \mathcal{L}(\mathbf{z}_{\mathrm{s}}, \mathbf{z}_{\mathrm{eff}})| \geq \mu |\mathbf{z}_{\mathrm{s}} - \mathbf{z}_{\mathrm{eff}}|, \tag{S2}$$

then the convergence of distillation loss can be guaranteed. A similar proof of convergence is used in Theorem A. 3 of [4] for linear distillation.

## S2   Test Error on CIFAR10 with ResNet

Figure S1:  Test error of knowledge distillation on CIFAR10 dataset with ResNet structure with respect to difference sample size $n$.

The settings of Fig.S1 are same as that of Fig.5 left in the main paper. All data points are averages of 5 times of training, for the purpose of eliminating variations in randomness. The curves shows a near power law relation of test error with respect to sample size $n$, especially when $n$ is large.

## S3 Proof of Transfer Risk Bound

One difference of our work from [4] is that the decision boundary contains the initial function as a bias term in NTK linearization $h(x) = 1 \Leftrightarrow f(x; w_0) + \Delta_w^\top \phi(x) > 0$. We use zero function weight change $\Delta_{w_z}$ to merge this bias term into part of weight change. Now we restate our risk bound and provide our proof.

**Theorem 1.** *Given input distribution $P(x)$, training samples $\mathbf{X} = [x_1, \cdots, x_n]$, oracle weight change $\Delta_{w_*}$, zero weight change $\Delta_{w_z}$ and accumulative angle distribution $p(\beta)$, the transfer risk is bounded by,*

$$R \le p(\frac{\pi}{2} - \bar{\alpha}_n), \tag{S3}$$

*where $\bar{\alpha}_n = \bar{\alpha}(\Delta_{w_*} - \Delta_{w_z}, \Delta_{\hat{w}} - \Delta_{w_z})$ and $\Delta_{\hat{w}}$ is student's converged weight change.*

*Proof.* Denote the oracle weight change and zero weight change as $\Delta_{w_*}$ and $\Delta_{w_z}$. Further we denote $\alpha(a, b) = \cos^{-1}(x^\top y / \sqrt{x^\top x \cdot y^\top y}) \in [0, \pi]$ as a supplement of $\bar{\alpha}$. The transfer risk can be written as

$$\begin{aligned}
\mathcal{R} &= \mathbb{P}_{x \sim P(x)} [(\Delta_{w_*} - \Delta_{w_z})^\top \phi(x) \cdot (\Delta_{\hat{w}} - \Delta_{w_z})^\top \phi(x) < 0] \\
&= \mathbb{P}_{x \sim P(x)} \left[ \alpha(\phi(x), \Delta_{w_*} - \Delta_{w_z}) < \frac{\pi}{2}, \alpha(\phi(x), \Delta_{\hat{w}} - \Delta_{w_z}) > \frac{\pi}{2} \right] \\
&+ \mathbb{P}_{x \sim P(x)} \left[ \alpha(\phi(x), \Delta_{w_*} - \Delta_{w_z}) > \frac{\pi}{2}, \alpha(\phi(x), \Delta_{\hat{w}} - \Delta_{w_z}) < \frac{\pi}{2} \right].
\end{aligned} \tag{S4}$$

We further assume $\alpha_n \le \pi/2$ so that $\alpha_n = \bar{\alpha}_n$. In actual network where $||\Delta_{w_z}||_2 \ll ||\Delta_{w_*}||_2$, this assumption is reasonable, since $\cos \alpha_n \propto \Delta_{\mathbf{z}}^\top \Theta_n^{-1} \Delta_{\mathbf{z}} > 0$. With the help of triangle inequality, $\alpha(a, b) \le \alpha(b, c) + \alpha(c, a)$, we can find a "easy" region where inputs are guaranteed to be correctly classified by student's model. For the case of $\alpha(\phi(x), \Delta_{w_*} - \Delta_{w_z}) < \pi/2$, if we assume the "easy" region is $\alpha(\phi(x), \Delta_{w_*} - \Delta_{w_z}) < \frac{\pi}{2} - \bar{\alpha}_n$, then

$$\begin{aligned}
\alpha(\phi(x), \Delta_{\hat{w}} - \Delta_{w_z}) &\le \alpha(\phi(x), \Delta_{w_*} - \Delta_{w_z}) + \alpha(\Delta_{w_*} - \Delta_{w_z}, \Delta_{\hat{w}} - \Delta_{w_z}) \\
&\le \frac{\pi}{2} - \bar{\alpha}_n + \bar{\alpha}_n = \frac{\pi}{2},
\end{aligned} \tag{S5}$$

which means student model also gives a correct prediction. Similarly, $\alpha(-a, b) \le \alpha(b, c) + \alpha(c, -a) \Rightarrow \pi - \alpha(a, b) \le \alpha(b, c) + \pi - \alpha(c, a)$. For the case of $\alpha(\phi(x), \Delta_{w_*} - \Delta_{w_z}) > \pi/2$, if we assume the "easy" region is $\pi - \alpha(\phi(x), \Delta_{w_*} - \Delta_{w_z}) < \frac{\pi}{2} - \bar{\alpha}_n$, then

$$\begin{aligned}
\pi - \alpha(\phi(x), \Delta_{\hat{w}} - \Delta_{w_z}) &\le \pi - \alpha(\phi(x), \Delta_{w_*} - \Delta_{w_z}) + \alpha(\Delta_{w_*} - \Delta_{w_z}, \Delta_{\hat{w}} - \Delta_{w_z}) \\
&\le \frac{\pi}{2} - \bar{\alpha}_n + \bar{\alpha}_n = \frac{\pi}{2}.
\end{aligned} \tag{S6}$$

Then we bound can the transfer risk by the worst case where all $\phi(x)$ outside this "easy" region is incorrectly classified, so that

$$\begin{aligned}
\mathcal{R} &\le \mathbb{P}_{x \sim P(x)} \left[ \frac{\pi}{2} > \alpha(\phi(x), \Delta_{w_*} - \Delta_{w_z}) > \frac{\pi}{2} - \bar{\alpha}_n \right] \\
&+ \mathbb{P}_{x \sim P(x)} \left[ \frac{\pi}{2} < \alpha(\phi(x), \Delta_{w_*} - \Delta_{w_z}) < \frac{\pi}{2} + \bar{\alpha}_n \right] \\
&= \mathbb{P}_{x \sim P(x)} \left[ \bar{\alpha}(\phi(x), \Delta_{w_*} - \Delta_{w_z}) > \frac{\pi}{2} - \bar{\alpha}_n \right] = p(\frac{\pi}{2} - \bar{\alpha}_n).
\end{aligned} \tag{S7}$$

$\square$

## S4 Experiment Details

All of our experiments are performed on `Pytorch 1.5`.

**Effective Student Logits Calculation** $z_{\text{s,eff}}(z_{\text{t}}, y_{\text{g}})$ is attained by solving

$$\lim_{\tau \to \infty} z_{\text{s}}(\tau) = \hat{z}_{\text{s}}, \quad \frac{d\ell}{d\hat{z}_{\text{s}}} = \frac{\rho}{T}(\sigma(\hat{z}_{\text{s}}/T) - \sigma(z_{\text{t}}/T)) + (1-\rho)(\sigma(\hat{z}_{\text{s}}) - y_{\text{g}}) = 0.$$

We use first order gradient descent with a learning rate of 10.0 and 10,000 iterations to get good results. Second order method like Newton's method is approachable, but it fails when $|z_{\text{t}}|$ is big and the second order gradient vanishes.

**Network Structure** In our experiment, we denote input layer as $d$, and the number of hidden layers as $L$. We also set the network to have a fixed hidden layer width $m$. All networks use NTK parameterization and initialization (described in [2]), which has the following form,

$$h^1 = \sigma_W W^0 x / \sqrt{d} + \sigma_b b^0, x^1 = g(h^1),$$
$$h^{l+1} = \sigma_W W^l x^l / \sqrt{m} + \sigma_b b^l, \; x^{l+1} = g(h^{l+1}), \; l = 1, 2, \cdots, L-1,$$
$$f(x; w) = \sigma_W W^L x^L / \sqrt{m} + \sigma_b b^L,$$

where $g(\cdot)$ is the activation function, $W_{ij}^l, b_i^l \sim \mathcal{N}(0,1)$ are the parameters of each layer, and $(\sigma_W, \sigma_b)$ are the hyperparameters and we use $\sigma_W = \sigma_b = 1.0$ through out our experiments. Further we denote $w = \cup_{l=0}^{L}\{W^l, b^l\}$ as the set of parameters.

The linearized network of random features is calculated according to $f_{\text{lin}}(x) = f(x; w_0) + \Delta_w^\top \partial_w f(x; w_0)$. Practically we calculate the derivative with the help of a `torch.autograd.functional.jvp` in `Pytorch 1.5`. All networks are optimized by standard Adam algorithm ($\beta_1 = 0.9, \beta_2 = 0.999$) with different learning rate $\eta$ and batch size $|\mathcal{D}|$.

**Neural Tangent Kernel(NTK) Calculation** NTK of a ReLU network is calculated according to Appendix C and E in [3].

**Input Distribution Design** In our experiments, the data distribution is fixed to be $\mathcal{N}(0, 5^2)$.

**Gaussian Mixture** The Gaussian mixture function in Fig.3 has the form of

$$z_{\text{gaussian}}(x) = \sum_{j=1}^{q} A_j \exp(-(x - x_j)^2 / \sigma_j)$$

$A_j, x_j, \sigma_j$ are all sampled with randomness, to insure the diversity of Gaussian mixture. $A_j$ are sampled around a fixed amplitude $A$, but with equal chance of positive and negative sign. $x_j$ is sampled according to a gaussian distribution $\mathcal{N}(0, \sigma_p)$ to make sure most of $x_j$ are in the distribution of $x$. $\sigma_j$ are also sampled around a fixed amplitude $\sigma$, but $\sigma$ is changed according to mode number $q$, $\sigma = 15/q^2$ so that all points can show their mode in $z_{\text{gaussian}}(x)$'s shape.

In the first difficulty control task (Fig.3, left), we control difficulty by control the number of modes $q$. In the second difficulty control task (Fig.3, right), $q$ is fixed, and we multiply $z_{\text{gaussian}}(x)$ with a random variable of $s \in \{\pm 1\}$, $z_*(x) = s \times z_{\text{gaussian}}(x)$. $s$ has probabilities of $\{1 - p_{\text{flip}}, p_{\text{flip}}\}$ to be $\{1, -1\}$.

In the following Fig. S2, we give a plot of Gaussian mixture function of different mode numbers.

**Calculation of Data Inefficiency** In the definition of data inefficiency $\mathcal{I}(n) = n[\ln \mathbb{E}||\Delta_{\hat{w},n+1}||_2 - \ln \mathbb{E}||\Delta_{\hat{w},n}||_2]$, we need to calculate $||\Delta_{\hat{w},n}||_2 = \sqrt{\Delta_{\mathbf{z}}^\top \Theta^{-1} \Delta_{\mathbf{z}}}$. $\Theta^{-1}\Delta_{\mathbf{z}}$ is calculated by the linear solver `torch.solve` in `Pytorch`.

### S4.1 Unstated Details of Each Figure

**Fig.2** The perfect teacher in Fig.2 of main text is the output of a network trained with hard labels generated by a gaussian mixture function. The teacher network has a setting of $d = 2, L = 5, m = 1024$. It is trained with learning rate $\eta = 0.0005$ and batch size $|D| = 4096$. We use *online-batch* training for all teachers, which means samples are regenerated after each epoch, in order to avoid the problem of sample correlation between teacher and student.

Figure S2: Examples of Gaussian mixture function. From left to right, each has a mode number of 10, 50, 250.

The students in Fig.2 of main text are the linearized network with a structure of $d = 2, L = 3, m = 2048$. They are trained with learning rate $\eta = 0.01$ and batch size $|D| = 512$. The distillation temperature is $T = 10.0$.

We also have to mention that, due to cross entropy loss, the convergence of student logits is especially hard when $|z_t| \gg 1$. To make our student more easily converge, and to make the scale of teacher match the scale of split generated by hard label $|z_t| \approx \Delta_{z,\text{split}}$, we reduce the scale of teacher's logits $z_{t,\text{new}} = r \times z_t$ by a factor $r$, called *reduction factor*. In our perfect distillation experiment(Fig.2), $r = 0.3$.

Here we give a plot of the decision boundary and output logits of the teacher in Fig. S3.

Figure S3: The decision boundary (left) and output logits(right) of teacher network.

**Fig.3** In Fig.3 of main text, calculation of $\sqrt{\Delta_{\mathbf{z}}^{\top} \Theta^{-1} \Delta_{\mathbf{z}}}$. $\Theta^{-1} \Delta_{\mathbf{z}}$ involves the output logits of initialized student network. The initialized network as a structure setting of $d = 2, L = 5, m = 1024$.

The data inefficiency is approximately a derivative $\mathcal{I}(n) = \partial \ln \mathbb{E}||\Delta_{\hat{w},n}||_2 / \partial \ln n$. For better illustration, we plot in Fig. S4 the intermediate quantity, $\mathbb{E}||\Delta_{\hat{w},n}||_2$ with respect to $n$.

**Fig.4** The teacher of Fig.4 has a network structure setting of $d = 2, L = 5, m = 1024$. It is trained by the hard label generated by the teacher network of Fig.2 with a learning rate $\eta = 0.01$ and a total epoch of 32768. In Fig.4 left we use teacher of different stopping epochs to give the plot, while in Fig.4 right we fix the teacher which stops at 511st epoch.

Here we also give plot of $\mathbb{E}||\Delta_{\hat{w},n}||_2$ with respect to $n$ in Fig. S5.

**Fig.5** **Left:** The teacher (ResNet50) has a test error of 6.97%. The students are all ResNet18 model, while the students' (ResNet18) baseline test error is 10.48% if trained from scratch. **Middle:** The

Figure S4: $||\Delta_{\hat{w}}||_2$ with respect to sample size $n$. In the plot $\Delta_{\mathbf{z}}$ of gaussian mixture functions are normalized by their scale $\Delta_{\mathbf{z},\text{new}} = \Delta_{\mathbf{z}}/||\Delta_{\mathbf{z}}||$. The dashed lines are references of random logits(upper dashed lines) and constant zero function(lower dashed lines).

Figure S5: $||\Delta_{\hat{w}}||_2$ with respect to sample size $n$. Similar to Fig. S4, $\Delta_{\mathbf{z}}$ are normalized.

teacher is trained by the hard label generated by the teacher network of Fig.2 with a learning rate $\eta = 0.0001$. It is early stopped at $e = 5113$ and at a test error of $1.06\%$ to make this phenomenon obvious. The sample size for student is $2^{14}$. **Right:** The teacher is trained in the same way as the Fig.5 middle, but with a total epoch of $10240$. In the calculation of $\langle \delta_{\hat{w}_{\text{h}}}, \Delta_{\hat{w}_{\text{c}}} \rangle$, a reduction factor of $r = 0.3$ is used.

## S5 Fitting a constant zero function.

This section is aimed to show that fitting a constant zero function is much easier to train than a normal task. In the following Fig. S6, we give plots on weight change $\Delta_w$ of fitting both constant zero function and the teacher function in Fig.2 using linearized network of structure $d = 2, L = 3, m = 2048$. The figure shows that constant zero function is faster to converge, and $\Delta_{w_{\text{z}}}$ is much smaller than $\Delta_w$ of a normal task.

## S6 Angle Bound of Random Features of ReLU Network

This section we aim to show that [4]'s bound $R_n \leq \min_\beta p(\beta) + p(\pi/2 - \beta)^n$ is loose in linearized wide neural network. Intuitively, $p(\beta) \to 1$ when $\beta \to 0$ and $p(\beta) \to 0$ when $\beta \to \pi/2$. If $p(\beta) < 1$ strictly when $\beta > 0$, then we can choose a $\beta \to \pi/2$ that has a relatively small $p(\beta)$, and with the help of the $n$ factor in $p(\pi/2 - \beta)^n$, the total risk bound can approach a small value.

However, as we will show below, the angle of random feature and oracle weight is bounded $|\cos \bar{\alpha}(\phi(x), \Delta_{w_*} - \Delta_{w_{\text{z}}})| \leq C_1$, so that $p(\beta) \equiv 1$ strictly for a range of $\beta \in [0, \beta_t], \beta_t \in (0, \pi/2)$. As we see in Fig.2 middle in the main text, this $\beta_t$ is probably near $\pi/2$, which means most random

Figure S6: Weight change plot with respect to training epoch. The bottom curve is that of training zero function, while other curves are distillation tasks, which are plotted as references.

feature $\phi(x)$ is nearly perpendicular to the oracle. Then the power factor of $p(\pi/2 - \beta)^n$ will not help reduce the risk bound, so that their risk bound gives an estimate of $R_n \geq 1$.

To demonstrate this, we use Eq. 7 in the main text to estimate the cosine of angle between random feature and oracle weight,

$$\cos \bar{\alpha}(\phi(x), \Delta_{w_*} - \Delta_{w_z}) = \frac{(\Delta_{w_*} - \Delta_{w_z})^T \phi(x)}{||\Delta_{w_*} - \Delta_{w_z}||_2 \cdot ||\phi(x)||_2} = \frac{z_{\mathrm{s,eff}}(x)}{||\Delta_{w_*} - \Delta_{w_z}||_2 \cdot \sqrt{\Theta(x,x)}}.$$

In the numerator, $z_{\mathrm{s,eff}}(x) \sim O(z_{\mathrm{t}}(x))$, especially when $|z_{\mathrm{t}}(x)| \gg 1$. In knowledge distillation, we assume teacher is also a ReLU network so that the output is also bounded by a linear function, $|z_{\mathrm{t}}(x))| \leq C_2 ||x||_2$. In the dominator, the factor $||\Delta_{w_*} - \Delta_{w_z}||_2$ is fixed and we find that for ReLU, the single value neural tangent kernel is $\Theta(x,x) \sim \Omega(x^\top x)$. Therefore this fraction is bounded $\cos \bar{\alpha}(\phi(x), \Delta_{w_*} - \Delta_{w_z}) \leq C_1 = C_2/||\Delta_{w_*} - \Delta_{w_z}||_2$. This $C_1$ is probably much smaller than 1, as we see in Fig.2 middle of the main text.

Figure S7: The NTK $\Theta(x,x)/(x^\top x)$ of a 3-hidden-layer network at different scale of x. We sampled 10000 input smaples according to $\mathcal{N}(0, \sigma_x^2)$, so that $\sigma_x$ denotes the scale of input. The figure shows that $\Theta(x,x) \geq x^\top x/4$ and the inequality tends to equality when the scale is large. This figure demonstrate that $\Theta(x,x) \sim \Omega(x^\top x)$.

## S7    Proof of Theorem 2

In this theorem we assume that $z_{\mathrm{s,eff}} \approx z_{\mathrm{t}} + (1 - \rho)\delta z_{\mathrm{h}}$, where $\delta z_{\mathrm{h}}$ is implicitly determined by hard labels. Then we can approximate

$$\langle \Delta_{\mathbf{z}_{\mathrm{g}}}, \Delta_{\mathbf{z}_{\mathrm{s,eff}}} \rangle_{\Theta_n} \approx \langle \Delta_{\mathbf{z}_{\mathrm{g}}}, \Delta_{z_{\mathrm{t}}} \rangle_{\Theta_n} + (1 - \rho)\langle \Delta_{\mathbf{z}_{\mathrm{g}}}, \Delta_{\delta z_{\mathrm{h}}} \rangle_{\Theta_n},$$

and

$$\frac{1}{\sqrt{\langle\Delta_{z_{\mathrm{s,eff}}},\Delta_{z_{\mathrm{s,eff}}}\rangle_{\Theta_n}}} \approx \frac{1}{\sqrt{\langle\Delta_{z_{\mathrm{t}}},\Delta_{z_{\mathrm{t}}}\rangle_{\Theta_n} + 2(1-\rho)\langle\Delta_{z_{\mathrm{t}}},\Delta_{\delta z_{\mathrm{h}}}\rangle_{\Theta_n}}}$$

$$\approx \frac{1}{\sqrt{\langle\Delta_{z_{\mathrm{t}}},\Delta_{z_{\mathrm{t}}}\rangle_{\Theta_n}}} - (1-\rho)\frac{\langle\Delta_{z_{\mathrm{t}}},\Delta_{\delta z_{\mathrm{h}}}\rangle_{\Theta_n}}{\langle\Delta_{z_{\mathrm{t}}},\Delta_{z_{\mathrm{t}}}\rangle_{\Theta_n}^{3/2}}.$$

By substituting the above two terms into Eq.11 of the main paper, and neglect higher order terms, then we can get Eq.12,

$$\left.\frac{\partial\cos\alpha(\Delta_{\hat{w}},\Delta_{w_{\mathrm{g}}})}{\partial(1-\rho)}\right|_{\rho=1} = \frac{1}{||\Delta_{w_{\mathrm{g}}}||_2\sqrt{\langle\Delta_{\mathbf{z}_{\mathrm{t}}},\Delta_{\mathbf{z}_{\mathrm{t}}}\rangle_{\Theta_n}}} \times$$
$$\left(\langle\Delta_{\mathbf{z}_{\mathrm{g}}},\delta\mathbf{z}_{\mathrm{h}}\rangle_{\Theta_n} - \frac{\langle\Delta_{\mathbf{z}_{\mathrm{g}}},\Delta_{\mathbf{z}_{\mathrm{t}}}\rangle_{\Theta_n}}{\langle\Delta_{\mathbf{z}_{\mathrm{t}}},\Delta_{\mathbf{z}_{\mathrm{t}}}\rangle_{\Theta_n}}\langle\Delta_{\mathbf{z}_{\mathrm{t}}},\delta\mathbf{z}_{\mathrm{h}}\rangle_{\Theta_n}\right). \tag{S8}$$