[Reviews · NeurIPS 2020]

Review 1

Summary and Contributions: This paper has a theoretical analysis on the knowledge distillation of a wide nerual network. In particular, they give a transfer risk bound, and discussed data inefficiency and imperfect teacher.

Strengths: The authors have done an in-depth analysis on the knowledge distillation. Further it is nice to see the discussion on the data inefficiency issue and imperfect teacher.

Weaknesses: As discussed in Section 3.2, the authors mentioned that the bound in [18] could show looseness, but the proposed bound could be loose with small sample size. Hence, it is hard to justify the advantages of the proposed analysis over the existing work [18]. Above Eq. (2), the authors assumed that the student is over-parameterized and wide. But recall that the motivation of studying student network is to have some lightweight network that can achieve a similar performance as that of the teacher network. The assumption of this paper can thus be questionable in practice. In Section 4, the authors had data inefficiency analysis. However if the sample size is small, the training of teacher network could be influenced as well. The analysis here then will become even complex.

Correctness: Most of the claims are right. But the assumption on the over-parameterized and wide student network could be rigorous in practice.

Clarity: The paper has been cleary written.

Relation to Prior Work: The authors have included the discussion of related works and explained the difference.

Reproducibility: Yes

Additional Feedback:


Review 2

Summary and Contributions: Knowledge distillation is a successful method of model compression and knowledge transfer. However, current knowledge distillation lacks a convincing theoretical understanding. This paper has theoretically analyzed the knowledge distillation of a wide neural network. First, a transfer risk bound is provided for the linearized model of the network. Then, a metric of the task’s training difficulty, (data ineffificiency) is proposed. Based on this metric, this paper finds that for a perfect teacher, a high ratio of teacher’s soft labels can be beneficial. Finally, for the case of an imperfect teacher, this paper finds that hard labels can correct the wrong prediction of the teacher, which explains the practice of mixing hard and soft labels.

Strengths: This paper has explained knowledge distillation in the setting of wide network linearization. This paper has provided a transfer risk bound based on angle distribution in random feature space. Early stopping of teacher and distillation with a higher soft ratio are both benefificial in making effificient use of data. Even if hard labels are data ineffificient, this paper has demonstrated that they can correct an imperfect teacher’s mistakes, and therefore a little portion of hard labels are needed in practical distillation.

Weaknesses: This paper investigated knowledge distillation in the setting of wide network linearization instead of a practical nonlinear neural network. There is no experiment on ImageNet. It is very hard to check the effectiveness of the proposed method.

Correctness: Following the logic of the paper, the analysis looks correct.

Clarity: The paper is understandable but it is not very straightforward.

Relation to Prior Work: [18] considers distillation of linear models and gives a loose transfer risk bound. The proposed method in this paper has improved the bound and generalize [18] to the case of linearization of an actual neural network.

Reproducibility: Yes

Additional Feedback: There is no experiment on ImageNet. It is very hard to check the effectiveness of the proposed method.


Review 3

Summary and Contributions: This paper gives a theoretical understanding of knowledge distillation, using a linear approximation of a general wide neural network for binary classification, as given by the neural tangent kernel. Assuming a perfect teacher, this work provides a tighter transfer risk bound than previous work, and for a linearized NN. Under the perfect teacher, they also provide a metric for the difficulty of the student to recover the weights of the teacher, under the number of samples in the training task. This metric is used to show that early stopping of the teacher improves data efficiency for the knowledge distillation task. Finally, allowing for an imperfect teacher, they show that the presence of hard (ground truth) labels can be helpful for correcting a wrong prediction (soft label) by the teacher, even as they worsen data efficiency. Hence, the recommendation is to use a few hard labels, as is standard practice in knowledge distillation. Overall, this work shows that the benefit of knowledge distillation comes from the ability of the teacher to provide the student network with a smoothened output function, but hard labels result in the loss of this smoothness, even as they promote better students.

Strengths: The submission presents useful results which help the theoretical understanding of knowledge distillation. Though there are no empirical results, the work seems to be of significance to the theoretical ML community. Moreover, it draws connections between KD and NTK, the latter being a recent, but impactful development. This paper is also one of the few in this conference that acknowledge the risk of negative impact of deep learning, if proper restrictions are not employed on its abuse.

Weaknesses: Since this paper is outside of my area of expertise, I can only comment on a couple of things I would have liked to see in this work, as follows: - Perhaps the inclusion of small scale synthetic experiments which empirically show the trade off of the performance benefit of hard labels vs. data efficiency would have made this work even stronger. - It is not clear what the practical implications of the data inefficiency metrics are; even though the conclusions from this metric seem to align well with findings from prior work.

Correctness: I am not very familiar with the theoretical background of neural tangent kernels, hence I'm not able to assess the correctness of bound presented in the paper.

Clarity: Yes.

Relation to Prior Work: Yes

Reproducibility: Yes

Additional Feedback: I would recommend a larger figure 1, even as the aspect ratio might need to change, since this figure is important to understanding this work. *Post author response* I thank the authors for providing a synthetic experiments, which makes this paper stronger. I stand by my original assessment of this paper.


Review 4

Summary and Contributions: Summary: The paper presents a theoretical analysis on knowledge distillation, where the student network is overparameterized so that it behaves as an NTK. In this case the paper proves a tighter transfer risk bound than a previous model in [18]. Then, the paper presents a metric to evaluate data efficiency in knowledge distillation and shows that early stopping / high soft ratio leads to higher efficiency. Finally, the paper analyzes the use of hard labels when the teacher network is not perfect.

Strengths: Pros: - The paper is well written and clear. The mathematical setup and analysis are solid. - The use of NTK is innovative in the area of knowledge distillation. - The risk bound is not based on (and better than) the classical Rademacher complexity, which is also novel. - The definition for data inefficiency is clear and makes a lot of sense. The result that higher soft ratio / early stopping can help obtain higher data efficiency is natural and important in explaining why they are useful. - The paper also talks about the use of hard labels for imperfect teacher networks, making the topic complete.

Weaknesses: Cons: - The paper assumes the student network f is overparameterized. However, there is no definition for f. There is also no comment on why you choose an overparameterized f (other than making it easier to analyze). In practice, student networks are usually small, which is mentioned in your introduction. Then, why is it meaningful to look at an overparameterized student network? More importantly, the paper assumes convergence when f is overparameterized, which is in fact not guaranteed. In [7], it is shown that overparameterized networks trained with sgd/gd can reach small l2 loss (because optimization is near convex). However, this may not be true for a completely different distillation loss. Therefore, it is necessary to prove convergence in for the distillation loss to make your assumption solid. - What is b at line 140? It seems the bound is tighter than classical bounds only when $b>1$. However, I don't see a definition for b. - As a theory paper, there is no formal theorem in both section 4 and section 5. It is necessary to extract the core ideas in these two sections into concise and rigorous theorems, even if they are expressed by words. - Finally, it is unclear how the three points made in this paper relate with each other. I don't see an overall picture after reading this paper; it seems like the three sections (3,4,5) are disjoint pieces and don't form a well-connected story. It would be clear if the authors can write a separate paragraph in introduction about how these points are related and together contribute to a bigger idea. There may be some points where I misunderstood or made mistakes. I am willing to increase my score if the above questions are resolved and made clear.

Correctness: Yes.

Clarity: Yes.

Relation to Prior Work: Yes.

Reproducibility: Yes

Additional Feedback:

[Author Response · NeurIPS 2020]

We thank all the reviewers for their precious suggestions. Here are our response.

*Reviewer 1*

**Q1: Advantages over bound in [18] under small sample size.** As our discussion in Sec 3.2 shows, [18]'s bound only
uses $\alpha_1'$, which is the angle between teacher and student's weight change when student is trained with *only one* sample.
Even in the regime of small sample size, $\alpha_n'$ in our bound (*considering all the training data*) is still much smaller than
$\alpha_1'$, therefore our bound still shows advantage.

**Q2: Reasons for considering over-parameterized student network.** It is well known that theoretical analysis
on practical distillation (i.e. small student network) is still a challenging open problem. The over-parameterization
assumption in this work is for the concern of theoretical convenience, and we acknowledge it might be a little impractical
at the current stage. In this work our focus is to establish an explicit and global relation between labels and network
parameters. NTK technique is one of the few choices so far. We believe our result is the first and an important step for
later conducting in-depth theoretical analysis on practical distillation.

**Q3: Small training sample size.** We fix the training sample size of teacher in our definition of data inefficiency, in
order to fix the difficulty of the task. Indeed, if teacher and student are trained by the same small sample set, the analysis
would be extremely complex. We will try to attack this problem as future work.

*Reviewer 2*

**Q1: No ImageNet experiments.** Training wide networks on ImageNet is extremely
We have validated our findings across synthetic and medium-scale(CIFAR10) datasets,
and we believe our findings still are true on large scale datasets.

*Reviewer 3*

**Q1: No small scale synthetic experiments.** We perform imperfect teacher distilla-
tion on synthetic dataset. The right figure shows a trade-off behavior between soft and
hard labels, which is similar to that of CIFAR10 dataset(Fig.5 left). We will add the
code and more details to future version of this paper.

**Q2: Practical impact of data inefficiency metric.** We already use data inefficiency
to show that early stopping and higher soft ratio may benefit distillation. In practice
we can further design new loss that has lower inefficiency to improve distillation.
More interestingly, data inefficiency can be a measure of the difficulty of certain task.

**Imperfect teacher distillation on synthetic dataset.** For teacher we use the training strategy in Fig.5 right. It is early stopped at $e = 5113$ and at a test error of $1.06\%$ to make this phenomenon obvious. The sample size for student is $2^{14}$. This figure shows a clear trade-off on soft ratio.

*Reviewer 4*

**Q1.1: Definition of neural network $f$.** It is stated in Supplementary Material Sec.S3.
We will add it into the main text of our paper.

**Q1.2: Reasons for over-parameterization of student.** We answer a similar question in **Q2** of *Reviewer 1*.

**Q1.3: Convergence of distillation loss.** Even though [7] only proves the convergence for only L2 loss, we believe this
also holds for our distillation loss, with a little bit of modification. The key idea of [7] is that, convergence is guaranteed
by the near constancy of NTK matrix $\hat{\Theta}(\mathbf{X}, \mathbf{X})$. Then we can use the following equation to prove convergence,
$\dot{\mathbf{z}}_s = -\eta\hat{\Theta}(\mathbf{X}, \mathbf{X})(\mathbf{z}_s - \mathbf{z}_{\text{eff}})$. As proved in the original paper of NTK([11]), the near constancy of $\hat{\Theta}(\mathbf{X}, \mathbf{X})$ has no
requirement on the type of loss, so this is also true for our distillation loss. Then, the difference only lies in the second
term $\mathbf{z}_s - \mathbf{z}_{\text{eff}}$, which in the case of distillation, is substituted with $\partial_{\mathbf{z}_s}\mathcal{L}(\mathbf{z}_s, \mathbf{z}_{\text{eff}})$ (same as eq.6 in [13]). Due to the
fact of finite training data and the convexity of $\mathcal{L}$ (w.r.t $\mathbf{z}_s$), the gradient can be lower bounded by another L2 loss,
so $|\partial_{\mathbf{z}_s}\mathcal{L}(\mathbf{z}_s, \mathbf{z}_{\text{eff}})| \geq \mu|\mathbf{z}_s - \mathbf{z}_{\text{eff}}|$, then the convergence of distillation loss can be guaranteed. A similar proof of
convergence is used in Theorem A.3 of [18] for linear distillation. We will add this point to the new version of the paper.

**Q2: Definition of $b$ in line 140.** Fig.4 middle and right suggest an empirical power law relation of data inefficiency
$\mathcal{I}(n)$ w.r.t. sample size $n$, $\mathcal{I}(n) \sim n^{-b}$. $b$ is the parameter to describe this relation. This observation help us to get the
asymptotic behavior of $||\Delta_{\hat{w}}||_2$ w.r.t. $n$, which leads to the asymptotic estimate of transfer risk bound. However, at
line 140 we haven't introduce $\mathcal{I}(n)$, therefore we give an equivalent definition of $\partial_n \ln ||\Delta_{\hat{w}}||_2 \sim n^{-b-1}$. We are not
certain whether $b > 1$ or not because $b$ depends on various hyper-parameters. However, we do find $b$ to be bigger when
teacher's stopping epoch is small (i.e. the task is easy), which might be better than classic bound.

**Q3: Lack of theorems in Sec.4, 5.** Thank you for the suggestion. We will summarize the core results and formulate
them into formal theorems.

**Q4: Organization of Sec.3, 4, 5.** Actually the three sections are not disjoint since each later section serves as a
complementary of previous one. In Sec.3 we use $p(\beta)$ as an estimate of transfer risk bound, but this measure needs
much effort to calculate and cannot show the obvious advantage of soft label. Therefore in Sec.4 we propose a simpler
measure of data inefficiency. Sec.5 serves as complementary of both Sec.3 and 4, for that the previous two sections
consider only perfect teacher.

[Meta-Review · NeurIPS 2020]

Knowledge distillation is not well understood and the reviewers agree that there's value in studying this topic. The results make wide network and NTK assumptions, which some reviewers (and this AC) eschew as unrealistic, however it's still exciting to see a theoretical step forward on such an opaque issue.